# Trends and between-Physician Variation in Laboratory Testing: A Retrospective Longitudinal Study in General Practice

**DOI:** 10.3390/jcm9061787

**Published:** 2020-06-08

**Authors:** Lisa D. Schumacher, Levy Jäger, Rahel Meier, Yael Rachamin, Oliver Senn, Thomas Rosemann, Stefan Markun

**Affiliations:** Institute of Primary Care, University and University Hospital Zurich, 8091 Zurich, Switzerland; lisadanielle.schumacher@usz.ch (L.D.S.); rahel.meier@usz.ch (R.M.); yael.rachamin@usz.ch (Y.R.); oliver.senn@usz.ch (O.S.); thomas.rosemann@usz.ch (T.R.); stefan.markun@usz.ch (S.M.)

**Keywords:** laboratory testing, trend, general practice, mixed-effect model, intraclass correlation coefficient

## Abstract

Laboratory tests are frequently ordered by general practitioners (GPs), but little is known about time trends and between-GP variation of their use. In this retrospective longitudinal study, we analyzed over six million consultations by Swiss GPs during the decade 2009–2018. For 15 commonly used test types, we defined specific laboratory testing rates (sLTR) as the percentage of consultations involving corresponding laboratory testing requests. Patient age- and sex-adjusted time trends of sLTR were modeled with mixed-effect logistic regression accounting for clustering of patients within GPs. We quantified between-GP variation by means of intraclass correlation coefficients (ICC). Nine out of the 15 laboratory test types considered showed significant temporal increases, most eminently vitamin D (ten-year odds ratio (OR) 1.88, 95% confidence interval (CI) 1.71–2.06) and glycated hemoglobin (ten-year OR 1.87, 95% CI 1.82–1.92). Test types both subject to substantial increase and high between-GP variation of sLTR were vitamin D (ICC 0.075), glycated hemoglobin (ICC 0.101), C-reactive protein (ICC 0.202), and vitamin B12 (ICC 0.166). Increasing testing frequencies and large between-GP variation of specific test type use pointed at inconsistencies of medical practice and potential overuse.

## 1. Introduction

Laboratory testing is one of the most frequently used diagnostic modalities in health care and influences up to 70% of all critical decisions [1]. During the past 20 years, the number of laboratory test types available for clinicians has doubled, and several studies have described an increase in the number of laboratory tests ordered in general practice [2,3,4,5,6]. The necessity of this increase in testing has been questioned, as healthcare systems are generally facing growing overuse of all kinds of diagnostic tests, also in primary care [7]. Furthermore, inappropriate testing might be detrimental to patients, causing psychological and physical harm as well as unnecessary financial burden [8].

Ideally, the use of laboratory tests should depend exclusively on patient factors that determine a clear indication. However, physician factors are also associated with the decision to order laboratory tests [9,10]. Physician sex [2], working environment [2], time since medical school graduation [11], tolerance of diagnostic uncertainty, and time pressure [9] have been found to be associated with test-ordering behavior. Some physicians assume that routine laboratory testing saves time, increases patient satisfaction and reassurance, and reduces the risk of malpractice liability [9,12]. In addition, fee-for-service billing in healthcare might contribute to overuse of laboratory testing by adding a financial incentive to increase quantity of testing [13]. In Swiss general practice, the availability of laboratory tests is high since their majority is reimbursed by mandatory health insurance and most general practitioners (GPs) maintain own facilities for point-of-care testing [14]. A subset of laboratory tests consists of reimbursable point-of-care tests, which poses additional direct financial incentives for getting the most of such on-site testing facilities [15]. Moreover, laboratory tests are a supply-sensitive element of care, meaning that increased availability may lead to overuse [16]. 

Even though the majority of laboratory tests in Switzerland are ordered in general practice [17], requesting patterns among GPs have not been the subject of recent research. With our study, we aimed to address this gap by describing trends and between-physician variation of laboratory testing in Swiss general practice.

## 2. Experimental Section

### 2.1. Study Design, Setting, and Participants

This study is a retrospective longitudinal database analysis in Swiss general practice using data from the Family Medicine ICPC (International Classification of Primary Care) Research using Electronic Medical Records (FIRE) project, which comprises of a network of GPs in German-speaking Switzerland exporting anonymized routine data from their electronic medical records to a central database. Among other information such as demographic patient characteristics, ICPC diagnostic codes, and drug prescriptions, GPs also contribute laboratory testing requests. We assessed data over one decade, ranging from the initiation of the FIRE project on 1 January 2009 to 31 December 2018. During this period, 389 GPs exported information on more than 6 million consultations of over 570,000 individual patients.

The local ethics committee of the canton of Zurich waived approval, as the project lay outside the scope of the Federal Act on Research involving Human Beings (BASEC-Nr. Req-2017-00797) [18].

### 2.2. Data Preparation and Selection

We validated laboratory test data by checking all exported labels, units, and distributions of test results for plausibility as well as for database errors (e.g., double counting because of multiple exports). The test panels complete blood count (CBC), urinalysis, liver enzymes, lipid profile, and electrolytes (sodium, chloride and potassium) were aggregated and considered as single tests. We approached data from all GPs in the FIRE project who exported at least 1000 consultations over the study period. Exclusion criteria were defined for the elimination of rare test types to increase the relevance of results and ensure sample sizes were sufficient for meaningful trend analyses and comparisons. Specifically, we excluded test types requested by less than 10% of all GPs over the observation period or requested by less than 10 GPs during any specific year. In addition, we excluded test types occurring with an among-GP median request rate below 1% of consultations over the study period. For each consultation, we extracted date, presence and type of laboratory test requests, (anonymized) IDs of patients and GPs as well as patient age and sex. Depending on the medical record software, some test types had not been exported by all GPs since their registration in the FIRE project, but export was enabled later during the study period after software updates. To assess testing behavior and frequencies, we therefore defined GP-specific observation starting points for counting both consultations and laboratory requests to fit the dates after which GPs actually started exporting respective test types.

### 2.3. Objectives

We aimed to determine GPs’ test type-specific usage frequencies of selected laboratory tests in terms of requesting rates per consultation and defined specific laboratory testing rates (sLTR) as the percentage of consultations by one single GP in which a specific laboratory test type was requested. Analyses of sLTR comprised of assessment of among-GP distribution, time trends, association with patients’ demographic factors, and measures of between-GP variation by extraction of the sLTR variance component attributable to within-GP clustering. For investigation of general testing variability and general association of testing with patient demographic factors, an overall laboratory testing rate (oLTR) was defined as the percentage of consultations in which at least one of the specific test types ultimately included in our analysis was requested.

### 2.4. Statistical Analysis

We approached sLTR analysis on the level of single consultations by definition of test type-specific binary variables denoting whether that particular laboratory test type was requested during a consultation. For oLTR, in an analogous manner, we introduced a binary variable encoding request during a consultation of at least one of the test types considered. For each test type separately, mixed-effect logistic regression was used to model the annual time trend with adjustment for patient sex and age (in years) as fixed factors. Random intercepts on the GP- and patient-level were introduced to account for repeated observations and clustering of patients within GPs. We determined odds ratios (OR) and corresponding 95% confidence intervals (95% CI) to report the effect of fixed factors on sLTR. Null models including time as the only fixed factor, but using the same random factor structure, provided a way to quantify the proportion of sLTR variance attributable to between-GP factors by assessing the corresponding intraclass correlation coefficients (ICC).

For analysis of oLTR, we fitted a mixed-effect model with hierarchical random intercepts on the GP- and patient-level with adjustments for patient age and sex as fixed factors to our data. A null random intercept-only model was used for computation of the GP-level ICC. As an additional measure of oLTR variation, we computed the central 90% range of GP-level OR for overall testing from the central 90% range of the GP-level random effect distribution as predicted by the null regression model. This quantity can be interpreted as the OR for requesting any laboratory test during a consultation of a given patient between a GP of relatively high oLTR (95th percentile) and a GP of relatively low oLTR (5th percentile) as predicted by the regression model.

We used R 3.6.3 (R Foundation for Statistical Computing, Vienna, Austria) for data cleaning and statistical analyses with the library lme4 for mixed-effect model fitting [19,20]. We reported statistical significance in terms of *p*-values using a significance threshold of 0.05.

## 3. Results

### 3.1. Selection Process

During the observation period (1 January 2009–31 December 2018), we approached 91 test types (*n* = 3,840,762 requests) out of which 21 were excluded (*n* = 53,879 or 1.4% of all requests) as they were requested by less than 10% of all GPs over the observation period or by less than 10 GPs during one specific year. Additionally, 54 test types (*n* = 338,362 or 8.8% of all requests) were excluded for occurring with an among-GP median sLTR below 1% of consultations. To enable meaningful time trend assessment, only test types present for at least five years in the database were considered, leading to exclusion of one test type (*n* = 8926 or 0.2% of all requests). In total, 6,116,587 consultations from 574,803 patients (52% female, median age at first consultation 44 years, interquartile range (IQR) 28–61 years) were analyzed after the exclusion of 221 patients (0.04% of total) due to missing information about age and/or sex (see Table 1 for characteristics of included patients). The 15 test types finally included (*n* = 3,435,297 or 89.4% of all requests) originated from 389 GPs working in 164 practices. Patients were followed over a median of 206 days (IQR 1–703 days) and were observed in a median number of four consultations (IQR 1–11 consultations). Appendix A summarizes the data selection process.

### 3.2. Test Type-Specific Use of Laboratory Tests

Crude among-GP distributions of sLTR before age and sex adjustment are visualized in Figure 1 and Figure 2 (overall and annual distributions, respectively) for the 15 test types addressed (distributions for the test types excluded from analysis can be found in Appendix A). The top three most frequently requested test types among GPs over the study period (Figure 1 and Figure 2) were complete blood count (among-GP median sLTR 15.1%, IQR 11.5–18.5%), C-reactive protein (CRP; among-GP median sLTR 10.4%, IQR 6.8–14.4%) and serum creatinine (among-GP median 7.2%, IQR 5.3–9.1%).

Test type-specific mixed-effect regression results are displayed graphically in Figure 3 (time trends in panel (**a**), associations with patient age in panel (**b**), associations with patient sex in panel (**c**), and GP-level ICCs in panel (**d**)). Numerical results can be found in the Appendix A. Of the 15 test types considered, nine showed a significant increase, four a significant decrease and two no significant time trend in terms of age- and sex-adjusted ten-year OR. We found the strongest increases for vitamin D (ten-year OR 1.88, 95% CI 1.71–2.06) and glycated hemoglobin (HbA1c; ten-year OR 1.87, 95% CI 1.82–1.92), and the strongest decreases for prothrombin time/international normalized ratio (PT/INR; ten-year OR 0.33, 95% CI 0.31–0.35) and erythrocyte sedimentation rate (ESR; ten-year OR 0.63, 95% CI 0.61–0.65).

Of the 15 test types analyzed, 12 were requested more frequently for increasing patient age, with the strongest effect for PT/INR (ten-year OR 1.49, 95% CI 1.47–1.52) and electrolytes (ten-year OR 1.30, 95% CI 1.30–1.31). On the other side, ferritin (ten-year OR 0.91, 95% CI 0.91–0.92), CRP (ten-year OR 0.94, 95% CI 0.93–0.94) and CBC (ten-year OR 0.99, 95% CI 0.98–0.99) were the only test types showing an increase of sLTR for decreasing patient age. Requesting of 14 test types was associated with patient sex. The strongest association with male sex was seen for lipid profile (male-to-female OR 1.61, 95% CI 1.58–1.64) and PT/INR (male-to-female OR 1.46, 95% CI 1.37–1.55), while the test types with strongest female sex association were ferritin (male-to-female OR 0.40, 95% CI 0.40–0.41) and vitamin D (male-to-female OR 0.54, 95% CI 0.52–0.55).

Concerning between-GP variation, we found the highest GP-level ICC for fasting glucose (0.231), CRP (0.202), and vitamin B12 (0.166), and the lowest for PT/INR (0.000), ferritin (0.018), and lipid profile (0.044).

### 3.3. Overall Use of Laboratory Tests

Non-adjusted among-GP median oLTR was 20.2% (IQR 16.9–24.0%) over the study period. In mixed-effect analysis, male patients were found to be tested less frequently than female patients (male-to-female OR 0.87, 95% CI 0.86–0.88), while increasing patient age was associated with higher oLTR (ten-year age OR 1.060, 95% CI 1.056–1.065). The GP-level ICC was obtained as 0.032, the central 90% GP-level OR range as 4.2 (5th log OR percentile −0.54, 95th log OR percentile 0.90). Figure 4 shows the distribution of GP-level random effect estimates.

## 4. Discussion

In this study, we analyzed more than three million single laboratory tests ordered by almost 400 Swiss GPs during the past decade. The most frequently requested test types were CBC, CRP, and renal function tests. Overall, the among-GP median testing rate amounted to 20% of consultations, but odds spanned over a four-fold ratio between low- and high-frequency testing GPs. Time trend analysis showed an increase of testing rates in two thirds of the included test types, especially vitamin D and HbA1c, which were subject to an almost two-fold increase over the past decade. Laboratory test types ranking high simultaneously in temporal increase and in between-GP variation were CRP, HbA1c, and vitamin B12.

We found increasing testing frequencies for most of the test types included in our study. Interestingly, our results mirrored findings from a comparable analysis by O’Sullivan et al. based on data from UK general practice gathered between the years 2000 and 2015 [4]. In both studies, there was an increase in requests of HbA1c, vitamin D, liver function tests, vitamin B12, CRP, ferritin, and thyroid function tests.

Vitamin D and HbA1c testing rates, however, almost doubled within the past decade and the extent of their increase sets them apart from other test types. Increases and potential overuse of vitamin D and HbA1c tests have been identified in other healthcare settings as well. The rise in HbA1c testing might be linked to its adoption for diagnosis of type 2 diabetes replacing blood glucose testing [21]. While we see no similarly compelling explanation for the increase of vitamin D testing, it has been speculated that intense media and individual interest might play a substantial role [22]. Still poorly understood, these trends are suspected to contribute to wasteful healthcare and to increase patient burden without introducing adequate benefits [22,23,24,25,26]. Increases in testing rates of CRP, ferritin, vitamin B12, electrolytes, and creatinine were also notable, but less pronounced.

Variation in laboratory testing was moderate on the level of overall testing. Most of the specific test types, however, were subject to variation exceeding an ICC of 0.05, which is unusual for measures in general practice [27]. Test types that were both associated with temporal increase and substantial variation between GPs were CRP, vitamin B12, HbA1c, and vitamin D. These test types are therefore most suspect for potential emerging overuse in Swiss general practice. Similar studies identified vitamin D and CRP as frequently ordered test types with relatively high between-physician variation in general practice, thereby also pointing at their potential overuse [28,29].

We used demographic variables primarily to adjust time trends, which was necessary to account for age and sex differences in individual GPs’ patient populations. However, several associations appeared and merit discussion. Increasing patient age was associated with higher requesting frequencies of most test types. This was unsurprising, as conditions requiring laboratory testing accumulate with increasing age [30]. Sex differences are, however, harder to interpret. Generally, we found that female patients received more laboratory testing compared to male patients, a result consistent with previous studies that found greater healthcare seeking behavior of female versus male patients in general practice [31,32]. Male sex, on the other hand, was associated with testing involved in cardiovascular risk estimation (lipid profile, HbA1c, fasting glucose). This may mirror the earlier manifestation of cardiovascular disease in in male patients [33]. However, this gender gap is closing [34] and, in addition, GPs are known to underestimate cardiovascular risks in female patients and tend to withhold preventative services to them [35]. Therefore, the sex difference we found may partly be a manifestation of an unwarranted gender gap. Female sex, on the other hand, was clearly associated with testing for vitamin D, ferritin, thyroid-stimulating hormone (TSH), and vitamin B12. Given the higher prevalence of osteoporosis [36], iron deficiency [37], and thyroid disorders [38] in female patients, our findings are concordant with epidemiologic disease distribution. The higher testing rate of vitamin B12 in female patients is less obvious to understand and may be linked to anemia investigations being more frequent in female patients due to iron deficiency and, in addition, to female sex being associated with vegetarian or vegan diet requiring vitamin B12 monitoring [39]. These factors, however, do not explain the high between-GP variation in vitamin B12 testing rates.

### Stengths and Limitations

A major strength of this study is its comprehensiveness in including 89% of single laboratory tests requested by GPs in a large and representative database from Swiss general practice. Trends, associations, and between-GP variation seem plausible and closely match results from comparable studies in the UK, adding to external validity of our research [4,29]. Lastly, to our best knowledge, this study is the first exploring between-GP variation of using different laboratory test types.

Our study presents several limitations. Firstly, we excluded rarely requested laboratory test types (those with <1% median among-physician sLTR) because they would have led to overdispersion and small sample sizes that would have been difficult to manage statistically and to interpret meaningfully. Secondly, younger GPs employed in urban and sub-urban areas are slightly over-represented in the FIRE database compared to national census [40]. Thirdly, the knowledge base in the domain of between-GP variation of laboratory use is scarce and we are unaware of other studies using ICCs for comparisons. Therefore, we must remain conservative with our interpretations of what constitutes unwarranted variation. On the other hand, however, our study contributes ICCs, which are notoriously difficult to estimate in advance, but are needed for planning potential future cluster-randomized trials aiming to reduce overuse of laboratory testing [27]. Lastly, this study was based on routine data collected from hundreds of GPs using different electronic medical records and export software. This made analysis vulnerable to mislabeled data, but we addressed this potential issue with due diligence and systematically double-checked plausibility of all laboratory test data in the database according to labels, units, and test result distributions on the level of each individual practice.

## 5. Conclusions

There is considerable between-GP variation of requesting laboratory tests, in part pointing at potential overuse. Laboratory test types associated with both high temporal increase and high between-GP variation were vitamin D, HbA1c, CRP, and vitamin B12. Our findings highlight the roadmap for initiatives aiming to better understand and ultimately reduce unwarranted variation and potential overuse of laboratory testing in general practice.

## Figures and Tables

**Figure 1 jcm-09-01787-f001:**
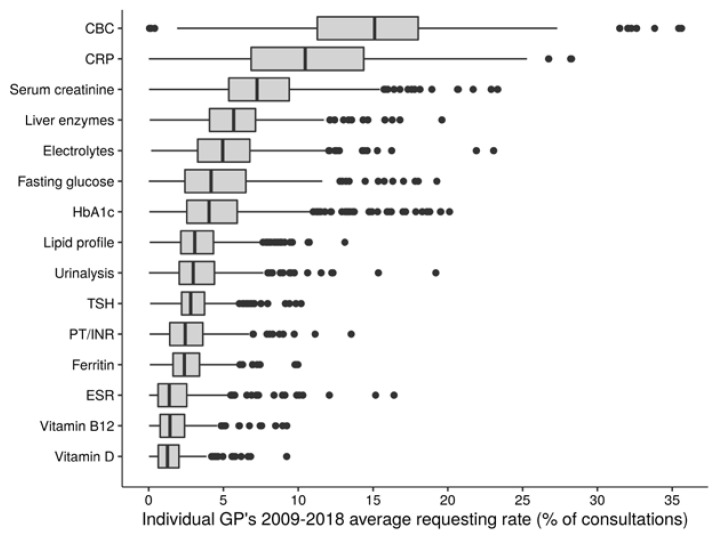
Crude among-general practitioner (GP) distributions of 2009–2018 average specific laboratory testing rates. Test types were included according to the criteria described in the main text. Type-specific laboratory testing rates were calculated for each GP as the percentage of consultations during the GP’s observation period involving a request of the respective test type. Abbreviations: CBC, complete blood count; CRP, C-reactive protein; ESR, erythrocyte sedimentation rate; HbA1c, glycated hemoglobin; PT/INR, prothrombin time/international normalized ratio; TSH, thyroid-stimulating hormone.

**Figure 2 jcm-09-01787-f002:**
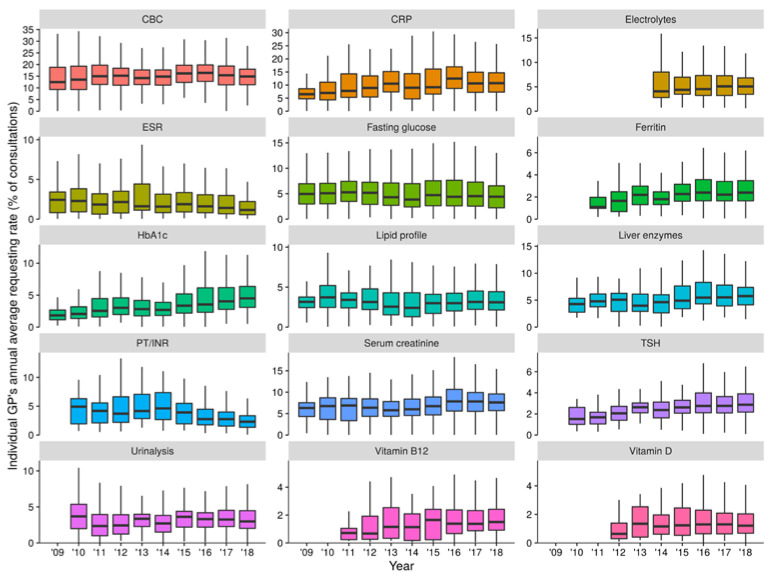
Crude among-general practitioner (GP) distributions of annual average specific laboratory testing rates for the years 2009–2018. Test types were included according to the criteria described in the main text. Outliers are omitted for better readability. Abbreviations: CBC, complete blood count; CRP, C-reactive protein; ESR, erythrocyte sedimentation rate; HbA1c, glycated hemoglobin; PT/INR, prothrombin time/international normalized ratio; TSH, thyroid-stimulating hormone.

**Figure 3 jcm-09-01787-f003:**
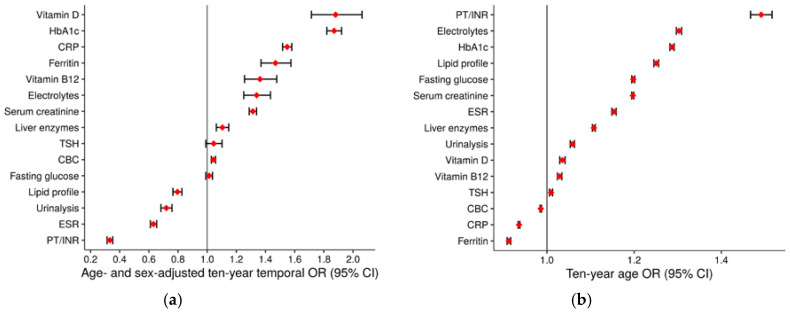
Results of mixed-effect regression analysis for specific laboratory testing rates. (**a**) Ten-year time trends. (**b**) Effect sizes of patient age. (**c**) Effect sizes of patient sex. (**d**) Between-general practitioner variance in terms of the null-model intraclass correlation coefficient. Abbreviations: CBC, complete blood count; CRP, C-reactive protein; ESR, erythrocyte sedimentation rate; HbA1c, glycated hemoglobin; PT/INR, prothrombin time/international normalized ratio; TSH, thyroid-stimulating hormone.

**Figure 4 jcm-09-01787-f004:**
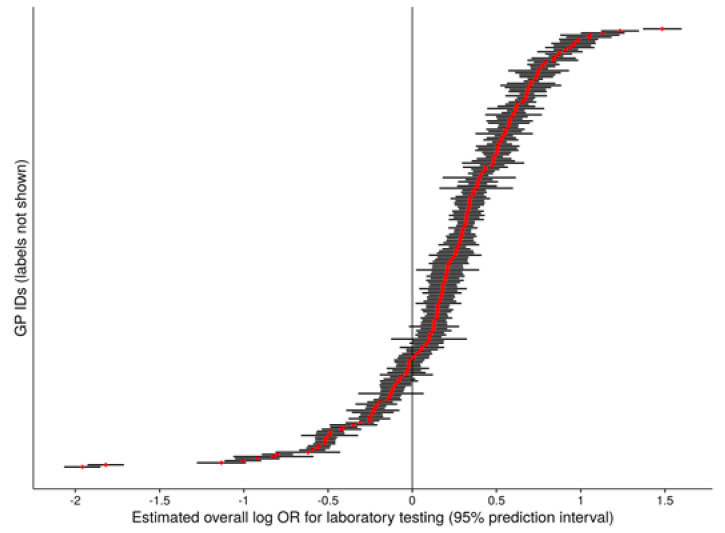
General practitioner-level random effect distribution. Each single point indicates the patient age- and sex-adjusted random effect estimate for one general practitioner (GP; *n* = 389) for the overall laboratory testing rate. Such a random effect estimate is numerically equivalent to the estimated log OR for laboratory testing during a consultation of a given patient by the corresponding GP relative to a rate given by the fixed intercept estimate of the null model (Table 2). The difference between the *x*-coordinates of any two point estimates can therefore be interpreted as the log OR between the corresponding GPs for laboratory testing during a consultation of a given patient.

**Table 1 jcm-09-01787-t001:** Characteristics of patients included for analysis.

Characteristic	At Least One Laboratory Test Reported (*n* = 315,807)	No Laboratory Tests Reported (*n* = 258,996)
Male sex, *n* (%)	172,810 (54.7)	132,062 (51.0)
Female sex, *n* (%)	142,997 (45.3)	126,934 (49.0)
Median age at observation start, years (IQR)	48 (32–64)	39 (25–56)
Median follow-up time, days (IQR)	406 (134–1152)	8 (1–227)
Median consultations per patient, *n* (IQR)	9 (1–19)	2 (1–4)

Abbreviations: IQR, interquartile range.

**Table 2 jcm-09-01787-t002:** Results of mixed-effect logistic regression analysis for overall laboratory testing.

**Full Model**				
Consultations, *n*	1,608,613			
**Fixed effects**	***β*** **(SE)**	**OR (95% CI)**	**Wald’s *χ*^2^**	***p*** **-Value**
Intercept	−1.95 (0.03)	0.14 (0.13–0.15)	−60	<0.001
Male sex	−0.143 (0.009)	0.87 (0.86–0.88)	−16	<0.001
Age (10 years)	0.058 (0.002)	1.060 (1.056–1.065)	27	<0.001
**Random effects**	**Variance estimate**	**Group members, *n***		
Patient ID	3.16	234,931		
GP ID	0.22	210		
**Null model**				
**Fixed effects**	***β*** **(SE)**	**OR (95% CI)**	**Wald’s *χ*^2^**	***p*** **-Value**
Intercept	−2.04 (0.03)	0.13 (0.12–0.14)	−72	<0.001
**Random effects**	**Variance estimate**	**ICC**		
Patient ID	3.17			
GP ID	0.21	0.032		

Abbreviations: *β*, coefficient estimate; SE, standard error; OR, odds ratio; CI, confidence interval.

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
