# Peer review of "Trends and between-Physician Variation in Laboratory Testing: A Retrospective Longitudinal Study in General Practice"

_jcm, 2020, doi:10.3390/jcm9061787_

Round 1

Reviewer 1 Report

This is a well conducted study providing important data. My only suggestion is that the authors could consider further developing the idea of using variability to detect inappropriate tests orders. This was discussed in a recent paper from my group:

Nguyen LT, Guo M, Hemmelgarn B, et al. Evaluating practice variance among family physicians to identify targets for laboratory utilization management. Clin Chim Acta. 2019;497:1‐5. doi:10.1016/j.cca.2019.06.017

Author Response

Point 1: This is a well conducted study providing important data. My only suggestion is that the authors could consider further developing the idea of using variability to detect inappropriate tests orders. This was discussed in a recent paper from my group:

Nguyen LT, Guo M, Hemmelgarn B, et al. Evaluating practice variance among family physicians to identify targets for laboratory utilization management. Clin Chim Acta. 2019;497:1‐5. doi:10.1016/j.cca.2019.06.017

Response 1: We thank the reviewer for appreciating our work. The reviewer pointed at an important article in the context of our study and suggested further elaboration of using variability measures for detection of inappropriate test orders. We had included a corresponding passage in the discussion section of the submitted manuscript version where we had related our findings to external sources and had implied a high between-physician variability to be an indicator of potential overuse (lines 228-232). We underlined this statement in the subsequent sentence and added the reference proposed by the reviewer:

Pages 7-8, lines 233-235: Similar studies identified vitamin D and CRP as frequently ordered test types with relatively high between-physician variation in general practice, thereby also pointing at their potential overuse [1,2].

References:

  1. Nguyen, L.T.; Guo, M.; Hemmelgarn, B.; Quan, H.; Clement, F.; Sajobi, T.; Thomas, R.; Turin, T.C.; Naugler, C. Evaluating practice variance among family physicians to identify targets for laboratory utilization management. Clinica chimica acta; international journal of clinical chemistry 2019, 497, 1-5.
  2. O'Sullivan, J.W.; Stevens, S.; Oke, J.; Hobbs, F.D.R.; Salisbury, C.; Little, P.; Goldacre, B.; Bankhead, C.; Aronson, J.K.; Heneghan, C., et al. Practice variation in the use of tests in UK primary care: a retrospective analysis of 16 million tests performed over 3.3 million patient years in 2015/16. BMC medicine 2018, 16, 229.

Reviewer 2 Report

General comments:

This paper used  robust methodology to study in trends in laboratory testing in general practice and found increases in test requesting over time, particularly for vitamin D and HbA1c, and between-GP variability in testing rates. The paper is well written and conclusions are clear. The Figures are appropriate and the strengths and weaknesses of the study are objectively described.

The results are of interest but perhaps unsurprising - temporal increases in Vit D and HbA1c in particular, between GP variability and sex differences in testing.  While it is well recognized that laboratory testing is increasing in many western healthcare systems , the reasons for this are unclear.  Nor is it clear what an optimal rate of testing should be.  Although the aims of this study have merit, a more interesting study might have considered other factors and determinants e.g. the attributes of GP practices [older v  more recently qualified GPs] , the extent to which testing frequency impacted on disease diagnosis, prevalence and outcome [ for example HbA1c], links between testing rates and other output measures such as prescribing rates, hospital referral rates etc [ there is some evidence that GPs who test more also prescribe more and make more hospital referrals] etc. The requesting of some tests relates clearly to clinical guidelines , for example during the time period of this study HbA1c may have been adopted as a diagnostic test for type 2 diabtes,  guidelines on the measurement of vitamin D  - the authors should comment on any relevant guidelines that came into use during the period of the  study.

Specific comments:

  1. In the Introduction the authors might briefly describe how laboratory testing is paid for and controlled in Switzerland - this is relevant to testing decisions.
  2. In the Introduction the authors might also mention that there may  detriments associated with over testing in addition to inappropriate use of resources - eg patient anxiety, false positive test results requiring further investigation, etc
  3. The authors might indicate whether there were any national or other guidelines issued over the observation period that might have impacted on testing rates for individual tests, in particular vitamin D and HbA1c Over the observation period the latter became a test for diagnosis of T2DM as well as a monitoring test]. There may also have been guidelines on cardiovascular risk management
  4. As outlined in my General comments above it would be interesting to relate testing data to clinical outcome data e.g. practice prevalence of diabetes, degree of HbA1c target attainment for diabetes patients,  degree of TSH target attainment for patients with hypothyroidism but I appreciate that this information may not be accessible to the authors. [I am aware of a small UK study which did not find any link between testing rates and quality of care outcome measures  (QOF)].

Author Response

General point: This paper used  robust methodology to study in trends in laboratory testing in general practice and found increases in test requesting over time, particularly for vitamin D and HbA1c, and between-GP variability in testing rates. The paper is well written and conclusions are clear. The Figures are appropriate and the strengths and weaknesses of the study are objectively described.

The results are of interest but perhaps unsurprising - temporal increases in Vit D and HbA1c in particular, between GP variability and sex differences in testing.  While it is well recognized that laboratory testing is increasing in many western healthcare systems, the reasons for this are unclear.  Nor is it clear what an optimal rate of testing should be.  Although the aims of this study have merit, a more interesting study might have considered other factors and determinants e.g. the attributes of GP practices [older v  more recently qualified GPs] , the extent to which testing frequency impacted on disease diagnosis, prevalence and outcome [ for example HbA1c], links between testing rates and other output measures such as prescribing rates, hospital referral rates etc [ there is some evidence that GPs who test more also prescribe more and make more hospital referrals] etc. The requesting of some tests relates clearly to clinical guidelines, for example during the time period of this study HbA1c may have been adopted as a diagnostic test for type 2 diabtes,  guidelines on the measurement of vitamin D  - the authors should comment on any relevant guidelines that came into use during the period of the  study.

General response: We thank the reviewer for the encouraging appreciation of our work and the constructive remarks, which we would like to address point-by-point:

Point 1: In the Introduction the authors might briefly describe how laboratory testing is paid for and controlled in Switzerland - this is relevant to testing decisions.

Response 1: The reviewer addresses healthcare system features relevant for testing decisions. We added corresponding information to the introduction including remuneration of testing in Switzerland and potential incentives for requesting:

Page 2, lines 47-51: In Swiss general practice, the availability of laboratory tests is high since their majority is reimbursed by mandatory health insurance and most general practitioners (GPs) maintain own facilities for point-of-care testing [3]. A subset of laboratory tests consists of reimbursable point-of-care tests, which poses additional direct financial incentives for getting the most of such on-site testing facilities [4].

Point 2: In the Introduction the authors might also mention that there may  detriments associated with over testing in addition to inappropriate use of resources - eg patient anxiety, false positive test results requiring further investigation, etc.

Response 2: The reviewer brings up the potential impact of laboratory test overuse on relevant patient outcomes. We added a corresponding sentence to the introduction citing a recent study from JAMA that fits the topic (and deleted the subsequence sentence as it would not be related to the previous one any longer and would not contribute any non-obvious information).

Page 1, lines 36-38, sentence added: Furthermore, inappropriate testing might be detrimental to patients, causing psychological and physical harm as well as unnecessary financial burden [5].

Page 1, lines 38-39, sentence deleted: However, a differentiated assessment is required, as increased testing might not necessarily concern all laboratory test types equally.

Point 3: The authors might indicate whether there were any national or other guidelines issued over the observation period that might have impacted on testing rates for individual tests, in particular vitamin D and HbA1c Over the observation period the latter became a test for diagnosis of T2DM as well as a monitoring test]. There may also have been guidelines on cardiovascular risk management

Response 3: The reviewer suggests elaborating on guidelines having potentially influenced testing behavior of Swiss GPs during the observation period. We feel it is of high relevance that there are no national institutes in Switzerland of uniformly high recognition among GPs. Numerous international institutes such as the USPSTF are recognized and supposedly influential, but the impact of each individual guideline on each individual test type can hardly be determined. For HbA1c, however, its adoption as a diagnostic test for type 2 diabetes during the observation period might indeed be a plausible explanation for the increase in testing, as pointed out by the reviewer. For vitamin D, we are unaware of a comparably compelling explanation. We elaborated these aspects accordingly by introducing the following sentences in the discussion:

Page 7, lines 221-224: The rise in HbA1c testing might be linked to its adoption for diagnosis of type 2 diabetes replacing blood glucose testing [6]. While we see no similarly compelling explanation for the increase of vitamin D testing, it has been speculated that intense media and individual interest might play a substantial role [7].

Point 4: As outlined in my General comments above it would be interesting to relate testing data to clinical outcome data e.g. practice prevalence of diabetes, degree of HbA1c target attainment for diabetes patients,  degree of TSH target attainment for patients with hypothyroidism but I appreciate that this information may not be accessible to the authors. [I am aware of a small UK study which did not find any link between testing rates and quality of care outcome measures  (QOF)].

Response 4: The reviewer points at interesting analyses providing deeper understanding of laboratory testing use by general practitioners. Indeed, we are planning according research projects, but anticipate considerable complexity requiring us to build specific frameworks for approaching each specific test type and clinical question. For example, we have already investigated feasibility of assessing appropriateness of vitamin D testing and supplementation including test results. We realized that various adjustments were necessary, e.g. for osteoporosis or risk factors for osteoporosis. In addition, we needed to identify concurrent vitamin D prescriptions and to make assumptions about the temporal relation between prescription and target achievement. We believe that this example shows how including such deeper analyses for each specific laboratory test type would lie beyond the scope of the present manuscript aiming at a comprehensive overview and comparison. Furthermore, additional measures such as hospitalization and referral rates are not available in our database.

References

3. Djalali, S.; Ursprung, N.; Rosemann, T.; Senn, O.; Tandjung, R. Undirected health IT implementation in ambulatory care favors paper-based workarounds and limits health data exchange. International journal of medical informatics 2015, 84, 920-932.

4. Federal Analysis List (Status as of 30 April 2020). Available online: https://www.bag.admin.ch/dam/bag/de/dokumente/kuv-leistungen/leistungen-und-tarife/Analysenliste/analysenliste-aenderungen-per-03-04-2020.pdf.download.pdf/Analysenliste%20%C3%84nderungen%20per%2030.04.2020.pdf (accessed on 04 June 2020)

5. Ganguli, I.; Simpkin, A.L.; Lupo, C.; Weissman, A.; Mainor, A.J.; Orav, E.J.; Rosenthal, M.B.; Colla, C.H.; Sequist, T.D. Cascades of Care After Incidental Findings in a US National Survey of Physicians. JAMA Netw Open 2019, 2, e1913325.

6. International Expert Committee report on the role of the A1C assay in the diagnosis of diabetes. Diabetes care 2009, 32, 1327-1334.

7. Rodd, C.; Sokoro, A.; Lix, L.M.; Thorlacius, L.; Moffatt, M.; Slater, J.; Bohm, E. Increased rates of 25-hydroxy vitamin D testing: Dissecting a modern epidemic. Clinical biochemistry 2018, 59, 56-61.